# Analysis of Barriers and Opportunities for Reclaimed Wastewater Use for Agriculture in Europe

**Enrique Mesa-Pérez**  and **Julio Berbel** *

Water, Environmental and Agricultural Resources Economics (WEARE), Universidad de Córdoba,
14001 Córdoba, Spain; emesa@uco.es
* Correspondence: berbel@uco.es

**Abstract:** This paper presents an analysis of the perception regarding reclaimed wastewater reuse in agriculture conducted in the European Union regions. The analysis is based upon a SWOT framework and applies a cluster analysis to reduce the dimension of the responses enabling an assessment of the different perceptions of water reuse. More than one hundred key actors identified among the regions participated in the evaluation of the relevance of aspects identified. The results indicate some groups of countries according to natural conditions (water scarcity) and the strategic role of agriculture as a key factor to determine agent's perceptions and attitudes. The results indicate that the forthcoming EU regulation of water reuse should focus in the problems of the perceived high cost of reclaimed water for farmers and the sanitary risk perception for irrigated crops by consumers as the critical points for fostering the use of reclaimed water in agriculture and the need for regional implementation of the global regulatory framework.

**Keywords:** water reuse; reclaimed water; SWOT analysis; cluster analysis

## 1. Introduction

Arid regions of the world usually have a demand for water that exceeds available resources. The use of reclaimed water is frequently mentioned as a "win–win" solution [1,2]. Previous experience in implementing reclaimed water for agricultural irrigation is satisfactory, especially in water-scarce areas [3], such as Spain, California, Australia [4], Jordan [5], or Italy [6]. Nevertheless, there are still some barriers and obstacles that should be reviewed by [7]. Therefore, water reuse "is considered vital to alleviate the demand on existing but limited water supplies and is gaining impetus throughout the world" [8], also as an alternative water resource to fight droughts and water scarcity [9].

Nevertheless, this opinion should be taken into consideration as wastewater is part of the hydrological cycle and its use in a closed basin where resources are already overallocated (as it is frequent in many regions) may increase exploitation of resources [10]. Additionally, the financial cost or the greenhouse gas emissions should also be considered. The main governance instrument in the EU is the Water Framework Directive (2000/60/EC) (WFD) [11]; the WFD has been successful slowing down the deterioration of water status and reducing (mainly point source) chemical pollution, regarding urban wastewater, 88% of EU wastewaters are subject to secondary treatment although water reuse is still low in the EU [12,13].

The EU included reclaimed water as part of the circular economy. As it is considered in the literature [14], the resources efficiency strategy and several regulations are developed with the aim to foster the use of reclaimed water. Water quantity and quality, including reclaimed water, is regulated by the EU mainly through the following: Water Framework Directive (2000/60/EC) [11], the Urban Waste Treatment Directive (91/271/ECC), the Scheme for Fertilizers (EC2003/2003) [15], or the Nitrates Directive (91/676/EEC) [16]. Closely related to EU water regulation is the Common Agricultural Policy

provisions 2014–2020 [17] and the Marine Directive [18]. Additionally, the EU also influences water reuse by strategic documents such as Commission communication on Water Scarcity and Droughts [19], Blueprint for Safeguarding European Waters [20], and the Circular Economy strategy [21]. Finally, several international initiatives like the Sustainable Development Goals included in the UN 2030 Agenda for Sustainable Development include fostering the use of reclaimed water within its goals.

However, the keystone in the implementation of reclaimed water for irrigation is the development of the "Regulation EU-2020/741 Minimum Requirements for Water Reuse" (European Commission, 2018) [21]. This regulation has been recently approved by the EU Parliament and seeks the homogenization of reclaimed water quality standards and water risk management systems for all the EU countries. There is a general agreement about water reuse brings benefits [22,23], but the proposed regulation should be adapted to varying conditions in each of the EU regions [2]. Consequently, a specific strategy should be used to foster reclaimed water in each region. This paper tries to answer this issue, analyzing the perception of the opportunities and barriers that several European regions face in the implementation of reclaimed water for agricultural irrigation.

This paper contributes to identifying regions with similar barriers and opportunities to implement reclaimed water in agriculture. We suggest that it is necessary for the implementation of specific strategies adapted to each regions' characteristics if a satisfactory reclaimed water implementation in agricultural irrigation is sought.

The paper continues as follows, firstly with the material and methods employed in the development of the research; secondly, with the cluster analysis; thirdly, with the results discussion; and finally, with the conclusions.

## 2. Materials and Methods

This research is based on the empirical work made during the European Project H2020 SUWANU-Europe [24], which proposes an exploratory analysis of the opportunities and barriers facing the use of reclaimed water in agriculture. To achieve it, this paper proposes a Cluster Analysis to know the similarities among the regions participating in the project: Belgium, Bulgaria, France, Germany, Greece, Italy, Portugal, and Spain. SUWANU-Europe departed from the results of the previous EU project (SUWANU) [19], which were used to support our analysis. The research design includes the survey of the relevant stakeholders (farmers, private sector, drinking water suppliers, wastewater suppliers, national and local administration, research institutions, and Non-Governmental Organizations (NGOs)) in the eight countries. In Appendix B is attached the table with the resume of key actors provided in the deliverable 2.1 of SUWANU [25].

### 2.1. Study Area

Regions included in the survey belong to eight European countries carefully selected to promote the adoption of water reuse strategies. The eight regions were selected following criteria of high technological development, Braunschweig; high water consumption in agriculture, Thessaloniki; high contribution of agriculture to regional economy, Andalusia and Plovdiv; total employment, Thessaloniki and Plovdiv; existing legislation, Andalusia; water stress, Thessaloniki, Tuscany, Antwerp, Limburg, and Andalusia; and high levels of rural population, Occitan, Santarem, Plovdiv, Thessaloniki, and Andalusia [2,4,24,26]. These regions belong to Belgium, Bulgaria, France, Germany, Greece, Italy, Portugal, and Spain. Table 1 illustrates the regional differences regarding urban wastewater treatment plants (WWTP) and related variables.

Regions under analysis differ in size and population. For that reason, we use data available in Table 1 such as the number of WWTP or the total discharge of wastewater allowed, to characterize reclaimed wastewater potential availability. Reclaimed water potential availability in these countries supports the idea of considering it as an alternative water resource, i.e., in some water abundant regions, such as Belgium the volume of treated water exceeds agriculture water demand.

**Table 1.** Data insight for regions.

| Country | BEL | BUL | FRA | GER | GRE | ITA | POR | ESP |
|---|---|---|---|---|---|---|---|---|
| Region | Antwerp and Limburg | Plovdiv | Occitanie | Braunschweig | Thessaloniki | Po-River | Alentejo | Andalusia |
| Number Urban WWTP | 108 | 1 | 3124 | 2 | 12 | 3579 | 103 | 668 |
| % Wastewater treated | 84% | 76% | 99% | 100% | n.a. | 82% | n.a. | 87% |
| Total discharge (hm$^3$) | 325.0 | 49.06 | 353.51 | 35.50 | 117.71 | n.a. | 36.09 | 698.17 |
| Reclaimed water use (hm$^3$/year) | 0.10 | 1.08 | 0.10 | 20.0 | 2.27 | n.a. | 30.88 | 41.42 |
| Irrigation demand (hm$^3$/year) | 15.79 | 186.0 | 1015.0 | n.a. | 1017.0 | 4750.0 | 512.58 | 4241.12 |
| % Wastewater treated | 84% | 76% | 99% | 100% | n.a. | 82% | n.a. | 87% |
| % Abstraction/ Resources (*) | 19% | 5% | 12% | 12% | 7% | 24% | 6% | 26% |

Source: SUWANU Europe Deliverable 1.1. resume table [27] and (*) Total water abstraction/Renewables resources. Data from EUROSTAT [28].

## 2.2. Material and Research Design

The material consists in the responses to a large survey conducted from May to July 2019, in the eight EU member states' regions. Aspects analyzed in the survey are categorized following the SWOT framework dimensions (Strength, Weakness, Opportunity, and Threat).

The proposed structure makes a more flexible comparison of aspects identified among the different regions for two main reasons. Firstly, not all regions have the same concern and expectations about the use of reclaimed water for irrigation. Consequently, aspects identified in each region can vary, making the comparison difficult. This classification respects those singularities and allows the aspects characterization following proposed categories. Secondly, whether all regions follow the same classification, the evaluation of the different aspects will show which categories received more attention in each region making results comparable.

Key actors were identified by the regional group from among members of all sectors related to the topic of reclaimed water in agriculture (policymakers, farmers' representatives, water technology companies, wastewater treatment suppliers, government institutions, and research institutions). Each one of the partners identified its regional key actors.

The identification of aspects involved in fostering reclaimed water for irrigation consisted of a three-step process. The first phase consisted in determining whether aspects identified in the previous EU project [29] were still relevant and proposing new aspects not included that could be relevant nowadays. Secondly, a design phase is conducted using different methods such as workshops, key actor interviews, and brief surveys to key actors. The aim of this phase is the final identification of all the aspects influencing reclaimed water implementation. Finally, the third step consisted in arranging the different aspects pointed in previous phases within SWOT framework dimensions (Strengths, Weakness, Opportunities, or Threats) and the categories explained in Figure 1. This process included a discussion about some results that considered an aspect as a strength or an opportunity at the same time, varying in relation to each key actor's opinion.

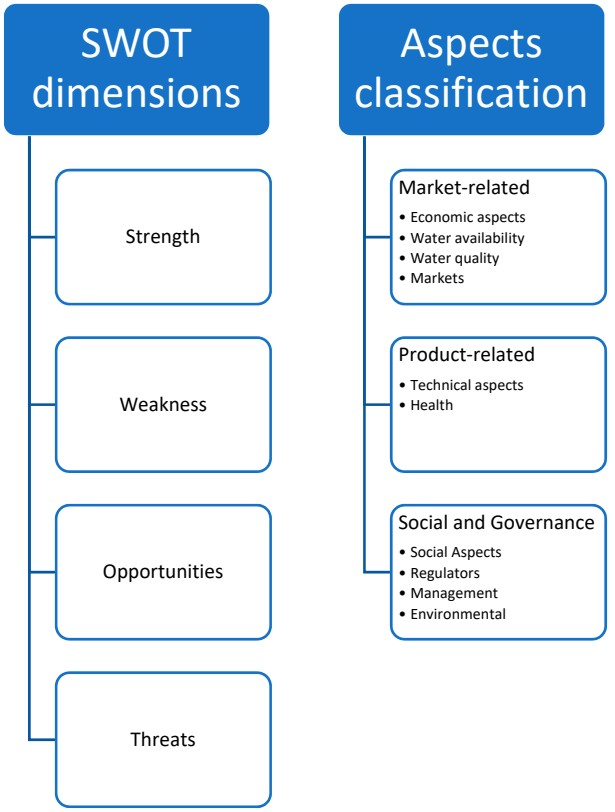

**Figure 1.** SWOT analysis dimensions and aspects classification proposed.

*2.3. Aspects Evaluation*

Although the use of SWOT analysis originated in business analysis, it also received uses outside this domain [9], and use of SWOT analysis to identify factors influencing the implementation of reclaimed water has already been made [8]. This paper focused on the evaluation to know the most relevant aspects influencing reclaimed water fostering for agricultural irrigation. The aim of this evaluation is the identification of the most relevant aspects in each region and the comparison of the results among the different regions. The classification proposed in Figure 1 will allow us to compare which groups of aspects have more relevance.

To evaluate aspects relevance, the methodology proposed is a Likert scale from 1 to 5. The Likert scale allows us to evaluate the agreement or disagreement for a series of statements [30,31] and is recommended the use of 5 levels (1 not relevant to 5 very important). This scale allows a neutral option, rate 3, for respondents without a clear answer about a question [31]. Most countries follow a 1–5 scale, although France and Germany use a scale 1–10 that was later converted to a 1–5 scale with the aim to compare results.

The methodology to evaluate aspects relevance also varies from one to another country. The most common tool used was an online survey sent to key users by email in Bulgaria, Greece, Italy, Portugal, and Spain. However, Belgium, France, and Germany evaluated the relevance of the different aspects surveying key actors directly during a workshop. Aspects identified and a preliminary analysis of the main results are available in SUWANU Europe Deliverable 2.1. [25]. In this research, we analyzed the compared results from the different regions following categories explained above (See Figure 1) trying to know which specific characteristics affect the implementation of reclaimed water as an alternative water resource.

## 3. Results

Generally, SWOT analysis makes a statistical description of the responses with an "expert opinion" for interpretation of the results. Our proposal is innovative as we will use cluster analysis to get some insight into the survey since we have eight countries with different objective characteristics (water scarcity, agricultural demand, etc.) and socioeconomic conditions.

Table 2 shows the results of the survey following the categories classification and SWOT dimensions explained in Figure 1. The higher the value, the more relevant is the aspect. For example, in Belgium, the most relevant categories are product-related strengths; in Bulgaria, strengths related with market-related issues; in France, market-related weaknesses and opportunities; in Germany, market-related opportunities; in Greece, issues about market-related strengths are the most relevant; in Italy, market-related strengths; in Portugal, market-related weaknesses; and in Spain, market-related strengths. This information will be analyzed more in detail following the cluster analysis results.

**Table 2.** Country average value for each for category for SWOT critera.

| Aspects Classification Following SWOT Dimensions | Belgium | Bulgaria | France | Germany | Greece | Italy | Portugal | Spain |
|---|---|---|---|---|---|---|---|---|
| Strength Market-related | 4.04 | **4.20** | 3.00 | 3.56 | **4.30** | **4.74** | **4.45** | **4.45** |
| Strength Product-related | **4.50** | **4.00** | 2.83 | **4.11** | **3.85** | 4.10 | 4.31 | **4.30** |
| Strength Social and Governance | 2.70 | 0.00 | **3.75** | 3.59 | 3.11 | **4.54** | 4.18 | **4.38** |
| Weakness Market-related | 4.23 | 3.67 | **3.83** | 3.08 | 3.06 | 4.20 | **4.94** | 3.48 |
| Weakness Product-related | **4.60** | **4.00** | 2.33 | 2.50 | 3.38 | 0.00 | **4.50** | 0.00 |
| Weakness Social and Governance | 3.73 | 3.25 | 2.25 | 3.50 | 3.25 | **4.56** | 3.86 | 3.20 |
| Opportunity Market-related | 4.05 | 3.50 | **3.83** | 4.50 | 2.79 | 3.92 | 3.61 | 3.82 |
| Opportunity Product-related | 4.00 | 3.00 | 2.50 | 3.72 | 3.42 | 4.21 | 3.42 | 0.00 |
| Opportunity Social and Governance | 4.06 | 2.92 | 2.38 | 3.42 | 3.21 | 3.69 | 3.63 | 4.01 |
| Threat Market-related | 3.80 | 3.25 | 3.50 | **3.75** | 3.28 | 3.35 | 3.70 | 3.63 |
| Threat Product-related | **4.30** | 3.00 | 1.50 | 3.08 | **4.00** | 0.00 | 4.42 | 0.00 |
| Threat Social and Governance | 3.95 | **4.11** | 3.38 | 3.33 | **4.51** | 4.17 | 3.71 | 4.03 |

Source: Own elaboration with data from SUWANU Europe SWOT Analysis. (1 means: no relevant; 5 means: very relevant).

This preliminary analysis shows that the perception of reclaimed water differs considerably according to each region's characteristics. We want to process this information and try to find similarities and differences that explain the perception of SWOT dimensions among the different regions to know the barriers and opportunities that reclaimed water is facing within each region. Consequently, this research drives a cluster analysis to evaluate which regions face similar barriers or opportunities in implementing reclaimed water for agricultural irrigation. For that reason, we simplify the results (see Table 3) to identify the type of barriers or opportunities the regions are facing. We calculated the average values of the aspects following the classification explained in Figure 1.

Table 3. Categories average evaluation.

| Aspects Classification | Belgium | Bulgaria | France | Germany | Greece | Italy | Portugal | Spain |
|---|---|---|---|---|---|---|---|---|
| Market-related | 4.03 | 3.66 | 3.54 | 3.72 | 3.36 | 4.05 | 4.18 | 3.85 |
| Product-related | 4.35 | 3.50 | 2.29 | 3.35 | 3.66 | 2.08 | 4.16 | 1.08 |
| Social and Governance | 3.61 | 2.57 | 2.94 | 3.46 | 3.52 | 4.24 | 3.85 | 3.91 |

Source: Own elaboration.

The analysis of agents' response is difficult to carry out based exclusively on descriptive statistics; therefore, we try some multivariate techniques whose primary purpose is to group objects based on the characteristics they possess. We select cluster analysis because it tries to identify internal homogeneity within the aspects of a group (cluster) and an external heterogeneity between each cluster [32].

We also analyze the differences among the regions following SWOT characteristics; on the one hand we pay attention to the prevalence of positive or negative aspects among the countries (see Table 4).

Table 4. Difference positive minus negative aspects SWOT analysis.

| Aspects Classification | Belgium | Bulgaria | France | Germany | Greece | Italy | Portugal | Spain |
|---|---|---|---|---|---|---|---|---|
| Market-related | 0.06 | 0.78 | −0.50 | 1.23 | 0.75 | 1.11 | −0.58 | 1.16 |
| Product-related | −0.40 | 0.00 | 1.50 | 2.25 | −0.11 | 8.31 | −1.19 | 4.30 |
| Social and Governance | −0.92 | −4.44 | 0.50 | 0.18 | −1.44 | −0.50 | 0.24 | 1.16 |

In this analysis, we can observe the prevalence of positive or negative aspects among the regions under analysis. On the one hand, Germany and Spain's key actors give more importance to positive issues in the three categories. On the other hand, Bulgaria, France, and Italy give a more positive relevance to two over three categories, and finally, Belgium, Greece, and Portugal give a higher negative relevance to two over three categories. This analysis could suggest that fostering reclaimed water could be "easier" in Spain or Germany than in Portugal or Bulgaria.

On the other hand, we provide an analysis of the prevalence of internal or external aspects among the countries. SWOT analysis evaluates internal aspects (strengths and weaknesses) and external aspects (opportunities and threats); consequently, we try to show which aspects are more relevant in each region. This analysis' results are provided in Table 5:

Table 5. Internal–external SWOT analysis.

| SWOT Aspects | Belgium | Bulgaria | France | Germany | Greece | Italy | Portugal | Spain |
|---|---|---|---|---|---|---|---|---|
| Int–Ext | −0.36 | −0.66 | 0.90 | −1.46 | −0.26 | 2.80 | 3.75 | 4.32 |

This analysis suggests that internal aspects are more relevant than external ones in France, Italy, Portugal, and Spain, while external aspects are more relevant in Belgium, Bulgaria, Germany, and Greece; we will discuss these results in the following part of the paper together with cluster analysis results.

Finally, cluster analysis is an exploratory data mining technique applied to the whole survey trying to force objects (responses in our case, regardless of the country of origin) to fall into the same group (called a cluster) following a similar definition of distance [32]. Our degrees of freedom "a priori" are eight countries by 12 groups: 4 SWOT dimensions × 3 categories (see Figure 1). We apply principal components analysis to reduce the dimensionality of the space of answers, although the results show that the Kaiser Meyer-Olkin (KMO) is lower to 0.6, recommending the use of original data [32]. Consequently, according to Hair [32], a logical basis is needed to determine the variables

to apply cluster. For that reason, this research concludes the proper variables are "market-related, product-related, and social and governance".

According to the results of cluster analysis (see Figure 2), we may identify two cluster groups: (a) Belgium, Portugal, Germany, Greece, and Bulgaria and (b) Italy, Spain, and France. The next section makes a deeper analysis of the perception in these four groups and tries to analyze results.

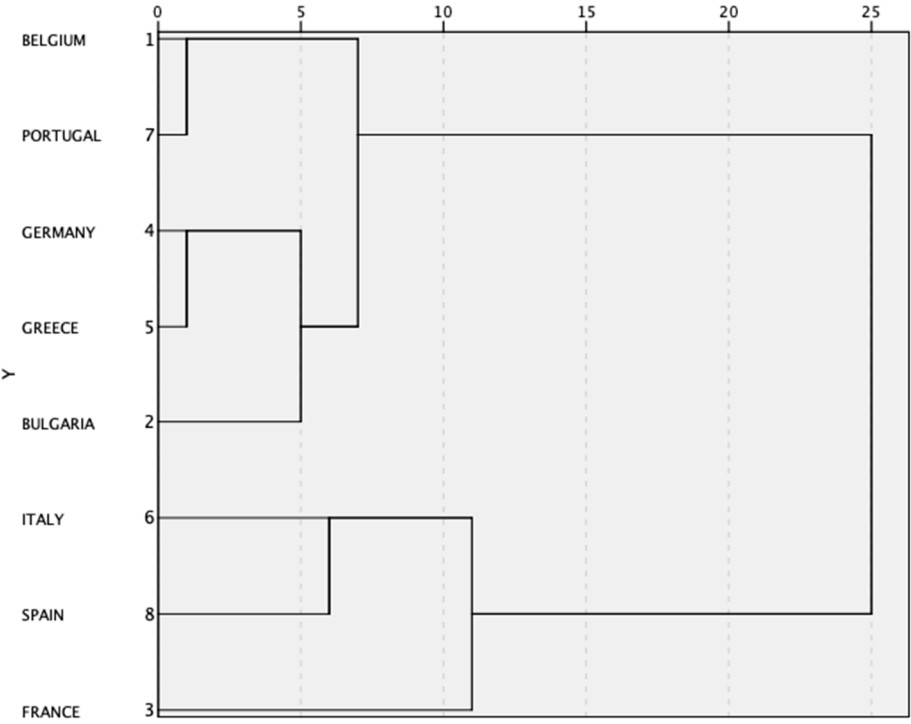

**Figure 2.** Cluster analysis result.

## 4. Discussion

This paper seeks similarities and differences among the barriers and opportunities perceived by key actors of eight EU regions. We conducted a SWOT analysis with the key actors' groups established for each one of the regions participating in the project. The first step was the identification of the relevant aspects. The SWOT analysis and the evaluation of the aspects were supported by a cluster analysis to identify similarities and differences among the regions. Following the categories proposed above (market-related, product-related, and social and governance), cluster analysis results in two groups: (a) Belgium, Bulgaria, Germany, Greece, and Portugal and (b) France, Italy, and Spain. An in-depth analysis of the aspects identified within the countries of each group is conducted.

Providing an in-depth analysis of the first cluster group (BE, BU, GE, GR, PT), we focus on the relevance of each category observed in Appendix A, Table A1 (see a resume in Table 6). We also provide a heatmap where the most relevant issues are colored red and the less green in Appendix A, Table A3. This first cluster key actors seem to agree about the high relevance of product-related issues. This can be a reflection of the potential use of reclaimed water supported in the existence of technological and technical conditions to treat wastewater (especially in Germany). However, in the same way, there exist some regions where product-related is also considered a weakness (Portugal and Bulgaria), or weakness and a threat (Belgium). Paying attention to the specific aspects identified by the key actors of those regions, we can identify risks for implementing reclaimed water, e.g., energy cost, the lack of infrastructure to distribute reclaimed water from the WWTP to the crops, or the necessity to learn from most advanced countries (see Cyprus and Israel). It can be observed a kind of consensus about the cost

of implementing water reuse and the cost of reclaimed water itself being aspects that should be faced by the public administration within these countries.

**Table 6.** Belgium, Bulgaria, Germany, Greece, and Portugal categories average evaluation.

| Aspects Classification | Portugal | Belgium | Bulgaria | Germany | Greece |
|---|---|---|---|---|---|
| Market-related | 4.18 | 4.03 | 3.66 | 3.72 | 3.36 |
| Product-related | 4.16 | 4.35 | 3.50 | 3.35 | 3.66 |
| Social and Governance | 3.85 | 3.61 | 2.57 | 3.46 | 3.52 |

Trying to understand how to face the cost management issues identified in Portugal, Belgium, or Bulgaria, we can observe that Germany shows just the opposite. German key actors may give higher relevance to product-related issues and market-related opportunities with comments such as "The potential self-financing business model of AV-BS (region of Braunschweig, Germany) where water fees paid by customers to support the system", or, the most relevant opportunity, "irrigation free of pollutants". These aspects are similar to the market-related issues identified in Belgium and Bulgaria, where the cost of reclaimed water for agricultural irrigation is considered a weakness. Moreover, these countries only identified one aspect as product-related strengths, e.g., "knowledge and technology about reclaimed water treatment". Consequently, as we explained just above, Belgium and Bulgaria give more relevance to product-related weaknesses than strengths. Nevertheless, the rest of the regions in this cluster, Germany, Greece, and Portugal, agreed in considering product-related aspects as a strength. These regions considered the existence of previous success stories and technology available, an issue that will facilitate the implementation of reclaimed water. However, it also seems relevant that product-related issues are considered as a threat in Portugal, Belgium, and Greece. In the case of Belgium, this is clear (see above), but in the case of Portugal and Greece, although these countries' key actors considered the existence of technology and technical conditions good to support reclaimed water implementation, they also suggested that the potential nanoparticles could require intensive treatment that threatens the use of reclaimed water. Besides, in the case of Portugal, the lack of infrastructure was not only considered a weakness but also a threat to overcome in the future.

It can be concluded that product-related issues are the most relevant in this cluster, positively such as in Germany or negatively like in the rest of the regions. The position regarding costs is the main difference between these regions. Paying attention to German product-related issues, they are considered the most relevant concerning strengths and opportunities dimensions. The technical experience of AV-SB (the German regional water company) in water reuse and the 4th wastewater treatment technology developed can be considered the solution for the high cost of reclaimed water that is perceived in the other countries. They have previous experience in reusing 20 hm$^3$ out 30 hm$^3$ wastewater discharge, and consequently, their cost is lower, but a relevant reason to understand this difference can be that Braunschweig is a small region, with only two WWTPs in comparison with the other, bigger regions with more WWTPs.

It can also be concluded that technology and technical issues to foster the use of reclaimed water for agriculture exist, and key actors within this cluster agreed about it. Nevertheless, energy costs or distribution costs should be overcome. Other aspects also received attention in this cluster. It can be observed how social and governance is considered a relevant threat in Belgium, Bulgaria, and Greece. On the one hand, Belgium and Bulgaria highlight that the new regulation will imply a high cost in implementing reclaimed water. On the other hand, Greece's key actors are more concerned about the public perception itself, e.g., "disagreement between various parties" or "uncertainty in the public … ". Portugal considered social and governance issues more a strength than a threat, e.g., their key actors highlight the existence of information programs and a perception of safety in using reclaimed water for agriculture. Finally, as explained above, Germany's key actors did not consider social and governance a relevant category, indeed one of the most relevant aspects identified is the no existence of water scarcity in the region.

Tables 4 and 5 illustrate the point that product-related issues are evaluated negatively in all the regions except for Germany, at the time that market-related issues are evaluated positively among the regions with the exception of Portugal (the most relevant category is market-related weakness, due to distribution costs). Being classified as market-related or product-related, this group is characterized by being concerned about the cost of implementing, distributing, and storing reclaimed water. In the case of Germany, the country is characterized by being able to drive this issue for the last years.

Regarding the second cluster, regions (FRA, ITA, ESP) give relevance to social and governance and market-related strengths (see Table 7). They perceive that the most relevant aspects are related to social and governance issues. This situation shows that society is concerned with water scarcity problems and considered that reclaimed water could help to fight it. Nevertheless, it is important to inform society properly, because threats about public perceptions also received higher attention, even when the new European Regulation implementation, the existence of reclamation standards, and good communication with users are considered a relevant strength to face the use of reclaimed water.

**Table 7.** France, Italy, and Spain categories average evaluation.

| Aspects Classification | France | Italy | Spain |
|:---:|:---:|:---:|:---:|
| Market-related | 3.54 | 4.05 | 3.85 |
| Product-related | 2.29 | 2.08 | 1.08 |
| Social and Governance | 2.94 | 4.24 | 3.91 |

In Appendix A, Table A2, it can be observed the evaluation of the different aspects' categories. France, Italy, and Spain give more relevance to internal than external aspects and they all agree to evaluate positively product-related issues (see Tables 4 and 5). It seems that key actors are optimistic about the implementation of reclaimed water in these regions. Paying attention to aspects identified as social and governance strength, the most common relevant category among this cluster, it can be observed that key actors considered the existence of an EU regulation such a quality guarantee to achieve public support. This characteristic opposes to the other cluster, where the EU regulation quality requirements were considered as an "extra cost". Besides, there exists an agreement about water scarcity and the necessity to seek alternative water sources. Consequently, the need for constant water flow for irrigation, the higher water demand for agricultural uses, and the existence of WWTP can lead to the consideration of reclaimed water as a proper alternative water resource. The difference between Greece and these countries can be motivated in the smaller number of WWTP and the greater availability of water regarding irrigated areas (see Table 1).

Finally, other aspects also received a higher score by key actors. For example, both Italy and Spain considered market-related issues as a strength. Aspects identified are related to the existence of quality standard, of constant water flow, or the environmentally friendly consideration of reclaimed water. All these aspects are related to the social and governance issues commented in the previous paragraph. In the case of France, market-related issues are considered an opportunity. The existence of big cities in the coastal areas and the increasing population support this evaluation. This aspect is also the most relevant in Germany, an issue that is supported by previous literature [2]. Finally, social and governance is also evaluated as a threat in France and Spain and as a weakness in Italy. In the case of France and Spain, the lack of a proper communication policy can result in consumers and wholesalers refusing to consume products irrigated with reclaimed water. The same happens in Italy, but in this case, the lack of public support is considered a weakness.

It can be concluded that this cluster is more optimistic than the first one. Although costs are also considered, more attention is paid to social and governance aspects. The motivation can result from a water scarcity situation and the higher water demand for agricultural irrigation. However, the need to communicate properly the benefits of irrigating with reclaimed water is also relevant for the environment and human health. For that reason, the new EU regulation is considered an opportunity

within these regions because it is considered a quality guarantee to avoid the distrust from consumers and food chain actors.

The groups that cluster analysis have shown can be seen as counterintuitive as they include only three southern countries (ES, FR, IT) meanwhile Greece and Portugal belong to the other cluster. The relative abundance of water in Portugal and the smaller amount of WWTP in Greece may be an explanation. Besides water abstraction (all uses) divided by available renewable resources in Portugal is closer to Northern countries than to neighboring Spain. Additionally, Italy and Spain have a competitive, export-oriented food industry, which may explain also the differentiation from other countries. Consequently, the relative water scarcity and the competitiveness of agribusiness may explain these results, although further research is required.

## 5. Conclusions

This paper provides an analysis that identifies the main opportunities and barriers faced by reclaimed water based upon cluster methodology and the interpretation of results. Although regions' hierarchy of topics varies, the global perception is that (a) high cost of reclaimed water for farmers and (b) social fear of products irrigated with reclaimed water should be the keystone of the EU strategy to foster the use of reclaimed water in agriculture.

In our research, we have detected that the perception of key actors varies according to the degree of water scarcity and the importance of irrigated agriculture. France, Italy, and Spain focus on water costs and the necessity to achieve consumer acceptance. Other countries without serious scarcity concerns focus on social governance issues to foster collaboration between farmers and the food chain. Policymakers should consider the impact of new EU regulation and support farmers in the financing of operation, at least in the initial stages, in order to strengthen the risk assurance system that will make transparency and social trust possible. Stronger involvement of regional or basin authorities will be probably the more efficient mechanism to promote water reuse avoiding farers and consumer resistance.

The analysis contributes to identifying the main barriers and opportunities that reclaimed water faces in its implementation process among the different regions. Consequently, when the European Commission seeks the approval of reclaimed water specific legislation, these differences should be considered. As this research concludes, not all the regions considered reclaimed water as an alternative water resource at the same level. In some cases, this is because the cost of water distribution is higher or maybe because there is not enough to achieve public support. Consequently, our opinion, based upon this evidence, is that there is a need for implementing different strategies in the different regions if a satisfactory reclaimed water implementation in agricultural irrigation is sought.

Further research could include other regions within the EU to obtain a complete landscape of reclaimed water barriers and opportunities.

**Author Contributions:** Both authors declare they worked equally in the development of this research. Both authors contributed to the development of the paper similarly. All authors have read and agreed to the published version of the manuscript.

**Funding:** This research is part of the EU project SUWANU-European, which is a Thematic Network funded by the EC under the H2020 program (contract number: 818088).

**Acknowledgments:** Authors want to acknowledge the support of participating experts and partners through all the Project SUWANU and assume the responsibility for all statements contained in this document.

**Conflicts of Interest:** The authors declare no conflict of interest.

## Appendix A. Tables SWOT Analysis Evaluation

**Table A1.** Belgium, Bulgaria, Germany, Greece, and Portugal aspects evaluation.

| Aspects Classification Following SWOT Dimensions | Belgium | Bulgaria | Germany | Greece | Portugal |
|---|---|---|---|---|---|
| Strength Market-related | 4.04 | 4.201 [1] | 3.56 | 4.302 [2] | 4.453 [3] |
| Strength Product-related | 4.502 | 4.003 [2] | 4.112 [3] | 3.85 | 4.31 |
| Strength Social and Governance | 2.70 | 0.00 | 3.59 | 3.11 | 4.18 |
| Weakness Market-related | 4.23 | 3.67 | 3.08 | 3.06 | 4.941 [1] |
| Weakness Product-related | 4.601 [1] | 4.003 [2] | 2.50 | 3.38 | 4.502 [2] |
| Weakness Social and Governance | 3.73 | 3.25 | 3.50 | 3.25 | 3.86 |
| Opportunity Market-related | 4.05 | 3.50 | 4.501 [2] | 2.79 | 3.61 |
| Opportunity Product-related | 4.00 | 3.00 | 3.72 | 3.42 | 3.42 |
| Opportunity Social and Governance | 4.06 | 2.92 | 3.42 | 3.21 | 3.63 |
| Threat Market-related | 3.80 | 3.25 | 3.753 [1] | 3.28 | 3.70 |
| Threat Product-related | 4.303 [3] | 3.00 | 3.08 | 4.003 [3] | 4.42 |
| Threat Social and Governance | 3.95 | 4.112 [3] | 3.33 | 4.511 [1] | 3.71 |

[1,2,3] represent the aspects with a higher relevance, according to key actors' evaluation per region.

**Table A2.** France, Italy, and Spain aspects evaluation.

| Aspects Classification Following SWOT Dimensions | France | Italy | Spain |
|---|---|---|---|
| Strength Market-related | 3.00 | 4.741 [1] | 4.451 [1] |
| Strength Product-related | 2.83 | 4.10 | 4.303 [3] |
| Strength Social and Governance | 3.752 [2] | 4.543 [3] | 4.382 [2] |
| Weakness Market-related | 3.831 [1] | 4.20 | 3.48 |
| Weakness Product-related | 2.33 | 0.00 | 0.00 |
| Weakness Social and Governance | 2.25 | 4.562 [2] | 3.20 |
| Opportunity Market-related | 3.831 [1] | 3.92 | 3.82 |
| Opportunity Product-related | 2.50 | 4.21 | 0.00 |
| Opportunity Social and Governance | 2.38 | 3.69 | 4.01 |
| Threat Market-related | 3.503 [3] | 3.35 | 3.63 |
| Threat Product-related | 1.50 | 0.00 | 0.00 |
| Threat Social and Governance | 3.38 | 4.17 | 4.03 |

[1,2,3] represent the aspects with a higher relevance, according to key actors' evaluation per region.

**Table A3.** Aspects Relevance Heatmap by Country.

| Aspects Classification Following SWOT Dimensions | Belgium | Bulgaria | France * | Germany | Greece | Italy * | Portugal | Spain * |
|---|---|---|---|---|---|---|---|---|
| Strength Market-related | 4.04 | 4.20 | 3.00 | 3.56 | 4.30 | 4.74 | 4.45 | 4.45 |
| Strength Product-related | 4.50 | 4.00 | 2.83 | 4.11 | 3.85 | 4.10 | 4.31 | 4.30 |
| Strength Social and Governance | 2.70 | 0.00 | 3.75 | 3.59 | 3.11 | 4.54 | 4.18 | 4.38 |
| Weakness Market-related | 4.23 | 3.67 | 3.83 | 3.08 | 3.06 | 4.20 | 4.94 | 3.48 |
| Weakness Product-related | 4.60 | 4.00 | 2.33 | 2.50 | 3.38 | 0.00 | 4.50 | 0.00 |
| Weakness Social and Governance | 3.73 | 3.25 | 2.25 | 3.50 | 3.25 | 4.56 | 3.86 | 3.20 |
| Opportunity Market-related | 4.05 | 3.50 | 3.83 | 4.50 | 2.79 | 3.92 | 3.61 | 3.82 |
| Opportunity Product-related | 4.00 | 3.00 | 2.50 | 3.72 | 3.42 | 4.21 | 3.42 | 0.00 |
| Opportunity Social and Governance | 4.06 | 2.92 | 2.38 | 3.42 | 3.21 | 3.69 | 3.63 | 4.01 |
| Threat Market-related | 3.80 | 3.25 | 3.50 | 3.75 | 3.28 | 3.35 | 3.70 | 3.63 |
| Threat Product-related | 4.30 | 3.00 | 1.50 | 3.08 | 4.00 | 0.00 | 4.42 | 0.00 |
| Threat Social and Governance | 3.95 | 4.11 | 3.38 | 3.33 | 4.51 | 4.17 | 3.71 | 4.03 |

* Countries belonging to cluster two. Colors represent the less relevant aspects (green) and the most relevant aspects (red), following the average relevance achieved in the survey.

## Appendix B

**Table A4.** Resume of key actors participating in the SWOT analysis, original from D2.1 SUWANU-Europe.

| Key Actors' Sector | Belgium | Bulgaria | France | Germany | Greece | Italy | Portugal | Spain |
|---|---|---|---|---|---|---|---|---|
| Farmers | 2 | 4 | | 3 | 3 | 8 | | 2 |
| Private Sector | 1 | 2 | | 1 | 4 | 3 | | 1 |
| Drinking water supplier | 1 | 1 | | | | | | |
| Wastewater supplier | 1 | 2 | 1 | 3 | | 1 | 2 | 5 |
| National administration | 1 | 2 | | | | | | 2 |
| Local administration | 2 | 4 | 1 | 2 | 3 | | | |
| Research institution | 1 | 2 | 1 | 1 | 10 | 3 | 4 | 8 |
| NGOs | | 1 | | | | 1 | | 6 |
| Total | 9 | 18 | 3 | 10 | 20 | 15 | 6 | 24 |

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
