# Peer review of "Analysis of Barriers and Opportunities for Reclaimed Wastewater Use for Agriculture in Europe"

_water, doi:10.3390/w12082308_

Round 1

Reviewer 1 Report

The paper studies the perception of reclaimed water in the agricultural sector. It is timely and offers a nuanced perspective on water reuse, which is often presented as a panacea to deal with water deficit. Especially, the SWOT analysis combined with a survey on perception carried out in 8 European regions enables the authors to put the implementation of reclaimed water into some context. To my knowledge, it is rare in the literature and highly relevant. The structure is good, and the paper fits with the aim & scope of water. I think there is a significant contribution to the literature. Nonetheless, I think there is for revisions before the paper suits for publication.

Please find below some suggestions that may help.

General Comments

I have two main comments. The first is a request for clarification in the methodology. The second is a suggestion of (minor) additional analysis that is likely to improve the paper significantly and broaden its readership.

1/ In table 2, there are several values below one, e.g., France's strength market is 0.75 or Bulgaria's strength social and governance is 0. As the Likert scale goes from 1 to 5, it is likely that some missing values have been coded 0 automatically when calculating the means. Besides, please the composition of the sample: how many actors have been surveyed, what are their characteristics?

2/ The mean of the SWOT variables (per category) is used to position and compare regions. This approach emphasizes the attention that is put on the theme, and I believe it is highly relevant. My suggestion is to offer a view of the perceived expected gains. A cost-benefits perspective is likely to provide complementary insights and a more comprehensive assessment of the perceptions in terms of decision-making. This cost-benefits perspective could simply be the difference between the mean of gains and the means of losses, e.g., for each region and category: (strength+opportunity)-(losses+threat). If it makes sense and it is feasible, I believe it could significantly improve the paper. Indeed, attention and expected benefits are strong drivers of policy and technological choices.

Minor comments.

The abstract is clear. I would add a short notice on the sample: the size and who are the agents analyzed to appraise the perception about wastewater reuse in agriculture

The introduction would benefit from a few additional words about the modernization of water use and governance in Europe. It could draw a broader picture situating the general interest of the paper.

l. 26: I think there is an error with the 5th reference.

l. 60-67: please clarify that the research design is multilevel with two different observation units, regions, and actors, and how the latter compounds the former.

Table 1: there is no available data about reclaimed water in the Po River. Reclaimed water is central to the paper, could provide any qualitative information in that case.

Table 2: change “treat” by “threat.” I wonder if turning the table into a heatmap would serve your purpose, i.e., identifying similarities and differences. Heatmaps are a fantastic tool in this regard. Also, it would be complimentary to the appendix that is well done and gathers detailed information.

To me, the results of the cluster analysis are counterintuitive. I would have expected a north/south divide that roughly reflects the hydroclimatic conditions and the difference of organization of the agricultural sector among regions. I recommend emphasizing that the results are somehow counterintuitive. It reinforces the relevance of carrying out this type of analysis.

The four last paragraphs of the discussion are of high added value. Moving them to the beginning of the section may help the reader.

Author Response

Reviewer 1

Dear reviewer, we are pleased to resubmit our article including the suggestions and comments made by the reviewers, whom we thank for their effort and time. The modifications in the text are marked (in red), and below you will find our replies to specific comments (in italics). We hope that this new version is now acceptable.

We would like to thank you very much for your comments. Answering your comments we identified a mistake in the calculation of the average values. We could solve it and we remade the whole analysis. As a result, the cluster analysis changed but we considered results are more coherent now than before.

We expect these changes would have improved the paper and you considered it capable of its publication.

Comments and Suggestions for Authors

The paper studies the perception of reclaimed water in the agricultural sector. It is timely and offers a nuanced perspective on water reuse, which is often presented as a panacea to deal with water deficit. Especially, the SWOT analysis combined with a survey on perception carried out in 8 European regions enables the authors to put the implementation of reclaimed water into some context. To my knowledge, it is rare in the literature and highly relevant. The structure is good, and the paper fits with the aim & scope of water. I think there is a significant contribution to the literature. Nonetheless, I think there is for revisions before the paper suits for publication.

Please find below some suggestions that may help.

General Comments

I have two main comments. The first is a request for clarification in the methodology. The second is a suggestion of (minor) additional analysis that is likely to improve the paper significantly and broaden its readership.

In table 2, there are several values below one, e.g., France's strength market is 0.75 or Bulgaria's strength social and governance is 0. As the Likert scale goes from 1 to 5, it is likely that some missing values have been coded 0 automatically when calculating the means.

We thank this comment to the reviewer because due to some categories do not include aspects in some countries, those categories were included with a zero. Which was wrong. To solve it we recalculate the means without zeros. Nevertheless, this changed our results, and we had to rewrite the whole discussion. We considered that our research now reflects a more accurate image of the regional situation.

Besides, please the composition of the sample: how many actors have been surveyed, what are their characteristics?

We introduce the sentence “The total number of actors surveyed has been 105 among the eight regions” in the abstract. In the methodological part, we included information about the number of actors involved per region: Belgium (9), Bulgaria (18), France (3), Germany (10), Greece (20), Italy (15), Portugal (6) and Spain (24) and we also included Appendix B with information about the numbers of key actors involved in the SWOT analysis per regions and sector they belong to.

We also include an Appendix B with the sector these key actors belong to. Any other information about this can be achieved in the “Deliverable 2.1. SWOT and PEST analysis”, available on https://suwanu-europe.eu/water-recycle-project-documents/

2/ The mean of the SWOT variables (per category) is used to position and compare regions. This approach emphasizes the attention that is put on the theme, and I believe it is highly relevant. My suggestion is to offer a view of the perceived expected gains. A cost-benefits perspective is likely to provide complementary insights and a more comprehensive assessment of the perceptions in terms of decision-making. This cost-benefits perspective could simply be the difference between the mean of gains and the means of losses, e.g., for each region and category: (strength opportunity)-(losses threat). If it makes sense and it is feasible, I believe it could significantly improve the paper. Indeed, attention and expected benefits are strong drivers of policy and technological choices.

Following your comment, we calculated the differences between:

(strengths + opportunities) – (weaknesses + threats) and the result is the following:

BELGIUM

BULGARIA

FRANCE

GERMANY

GREECE

ITALY

PORTUGAL

SPAIN

MR

0,06

0,78

-0,50

1,23

0,75

1,11

-0,58

1,16

PR

-0,40

0,00

1,50

2,25

-0,11

8,31

-1,19

4,30

SG

-0,92

-4,44

0,50

0,18

-1,44

-0,50

0,24

1,16

Besides, following SWOT nature, we calculated the difference between internal aspects (strengths and weaknesses) and external aspects (opportunities and threats), resulting:

BELGIUM

BULGARIA

FRANCE

GERMANY

GREECE

ITALY

PORTUGAL

SPAIN

Int-Ext

-0,36

-0,66

0,90

-1,46

-0,26

2,80

3,75

4,32

These analyses are included in section 3 and are useful to explain the difference between the new clusters.

Minor comments.

The abstract is clear. I would add a short notice on the sample: the size and who are the agents analysed to appraise the perception about wastewater reuse in agriculture

We have included this sentence in the abstract and Appendix B with more information:

More than one hundred key actors identified among the regions participated in the evaluation of the relevance of aspects identified

The introduction would benefit from a few additional words about the modernization of water use and governance in Europe. It could draw a broader picture situating the general interest of the paper.

We have included this sentence in the introduction:

The main governance instrument in the EU is the Water Framework Directive, and according to the (European Commission, 2019)  the WFD has been successful slowing down the deterioration of water status and reducing (mainly point source) chemical pollution, regarding urban wastewater, 88% of EU wastewater are subject to secondary treatment although water reuse is still low in the EU.

  1. 26: I think there is an error with the 5th reference.

We have double-checked the reference and do not find the error

  1. 60-67: please clarify that the research design is multilevel with two different observation units, regions, and actors, and how the latter compounds the former.

We have included modified the paragraph after line 65including this information.

The results from a previous EU project (SuWaNu) [19] were used to support our analysis. The research design includes a survey of the relevant stakeholders (farmers, operators, consumers, policymakers, experts) in the eight countries.

Table 1: there is no available data about reclaimed water in the Po River. Reclaimed water is central to the paper, could provide any qualitative information in that case.

We tried to include this information in the research, however, it has been impossible to find it out. Neither our Italian colleagues nor us could achieve the information. Nevertheless, we include additional information in Table 1.

Table 2: change “treat” by “threat.” I wonder if turning the table into a heatmap would serve your purpose, i.e., identifying similarities and differences. Heatmaps are a fantastic tool in this regard. Also, it would be complimentary to the appendix that is well done and gathers detailed information.

We have included the heatmap in appendix A

To me, the results of the cluster analysis are counterintuitive. I would have expected a north/south divide that roughly reflects the hydroclimatic conditions and the difference of organization of the agricultural sector among regions. I recommend emphasizing that the results are somehow counterintuitive. It reinforces the relevance of carrying out this type of analysis.

As you expected, new results suggest this division. More specifically, a division between the Mediterranean and high extension crop countries (Spain, France, and Italy; with the exception of Greece) and the rest of the regions.

The four last paragraphs of the discussion are of high added value. Moving them to the beginning of the section may help the reader.

We thank you for the suggestion and have modified the section accordingly

Reviewer 2 Report

The paper “Analysis of Barriers and Opportunities for Reclaimed Wastewater Use for Agriculture in Europe” is interesting and well-written. In my opinion it is not innovative from the methodological side, but it provides a useful comparison of the perception of reclaimed water reuse in different EU countries. I think it could be considered for publication once some issues are clarified.

The English should be carefully revised, there are a few typos and errors throughout the text.

Section 2.1 provides an interesting comparison among the multiple regions involved in the analysis, mainly focusing on the rationale for study areas selection. However, I feel that the analysis of similarities and differences should be performed more into details and in a more systematic way. For example, since the study is focused on problem ‘perception’, differences in readiness, awareness, and technological situation should be mentioned more into details. Probably this could be addressed also expanding Table 1.

Section 2.2. Additional details would be needed on the key actors identified and involved into the process. The description in lines 96-99 is too general, and more information should be included.

Did stakeholder have the same weight in the process?

Section 2.3. Were the aspects pre-selected or co-defined along with stakeholders?

Additional aspects would be needed on the methodological side: did the authors use only individual interviews? Was any workshop of group activity performed? Howe were the differences in the scores analyzed and considered in the results?

Not sure the Section 4 is a Discussion, because it is a critical analysis of the results. Indeed, this should be improved since it is the most important Section of the paper. Specifically, the Authors should try to go beyond the ‘numerical values’, trying to find the root causes of such differences (e.g. cultural, economic, technological, etc.). This should be the value added of the proposed work. Furthermore, I would also suggest also some additional reasoning on the potential measures to overcome the identified barriers, which should make this analysis useful for policy-and decision-makers.

Author Response

Reviewer 2

Dear reviewer, we are pleased to resubmit our article including the suggestions and comments made by the reviewers, whom we thank for their effort and time. The modifications in the text are marked (in red), and below you will find our replies to specific comments (in italics). We hope that this new version is now acceptable.

Thank you very much for your comments. We tried to apply them and improve the paper. However, we must comment you something that happened with reviewer 1 comments. R1 suggested some comments and applying them we found out a mistake in the calculation of the means values. Consequently, we solved the wrongdoing and we remade the analysis.

As a result, the cluster analysis changed and we identified two new clusters: the first one with Belgium, Bulgaria, Germany, Greece and Portugal, and a second one with France, Italy, and Spain. We conducted an in-depth analysis again to understand the structure of these clusters and a new discussion is provided.

In our opinion, results are now more coherent and there appears expected north-south differences. In this case, the Mediterranean and not Mediterranean countries (with the exception of Greece). We did not change nor the methodology, nor the study. Following R1 comments we also add two new analyses to section 3 about the positive-negative SWOT aspects prevalence and the internal-external SWOT prevalence.

In the following lines, we explained how we answered your comments.

Comments and Suggestions for Authors

The paper “Analysis of Barriers and Opportunities for Reclaimed Wastewater Use for Agriculture in Europe” is interesting and well-written. In my opinion it is not innovative from the methodological side, but it provides a useful comparison of the perception of reclaimed water reuse in different EU countries. I think it could be considered for publication once some issues are clarified.

Your comments are welcomed, and we are grateful for your time.

The English should be carefully revised, there are a few typos and errors throughout the text.

The original manuscript was revised by a professional service, nevertheless, we double-checked the new version after new changes were applied.

Section 2.1 provides an interesting comparison among the multiple regions involved in the analysis, mainly focusing on the rationale for study areas selection. However, I feel that the analysis of similarities and differences should be performed more into details and in a more systematic way. For example, since the study is focused on problem ‘perception’, differences in readiness, awareness, and technological situation should be mentioned more into details. Probably this could be addressed also expanding Table 1.

Table 1 has been expanded including additional information

Section 2.2. Additional details would be needed on the key actors identified and involved into the process. The description in lines 96-99 is too general, and more information should be included. Did stakeholder have the same weight in the process?

L99) (..) partners identified its regional key actors. We add information to the paragraph: Belgium (9), Bulgaria (18), France (3), Germany (10), Greece (20), Italy (15), Portugal (6) and Spain (24), in appendix B is available a table with information about the sector each key actors belong to per country.

Moreover, we included an appendix B with information about key actors involved per country and sector.

Section 2.3. Were the aspects pre-selected or co-defined along with stakeholders?

As we explain in line 105 and onwards, we departed from SUWANU project [19] SWOT analysis results. We took those results and together with key actors we re-evaluated their relevance and identified new aspects not included in the previous project. Maybe, this information is not clear, consequently, we rewrite the section.

Additional aspects would be needed on the methodological side: did the authors use only individual interviews? Was any workshop of group activity performed? Howe were the differences in the scores analyzed and considered in the results?

No, different methodologies were employed by each partner, we included additional information about it in point 2.3. Some regions conducted an online survey, and another consulted their key actors during their regional workshops. We took these differences into account.

Not sure the Section 4 is a Discussion, because it is a critical analysis of the results. Indeed, this should be improved since it is the most important Section of the paper. Specifically, the Authors should try to go beyond the ‘numerical values’, trying to find the root causes of such differences (e.g. cultural, economic, technological, etc.). This should be the value added of the proposed work.

We have modified the Discussion section, included some possible explanations of the findings.

Furthermore, I would also suggest also some additional reasoning on the potential measures to overcome the identified barriers, which should make this analysis useful for policy-and decision-makers.

We have included additional arguments in the conclusion regarding perceived barriers and solutions

Reviewer 3 Report

In this article an analysis is carried out to determine the existing perception of the use of reclaimed wastewater for agricultural irrigation in different regions of the European Union. SWOT analysis and cluster analysis are used for this purpose.

In general, the article is well-written, well-organized and all sections are well-developed. The research design is appropriate, and the method is adequately described. The result is clearly presented and the conclusion are supported by the result. However, the article should introduce some general and specific changes before publishing.

In order to improve the content of this paper the authors should consider following general suggestions:

  • In line 2, when it is mentioned that the use of reclaimed water is often considered a 'win-win' solution, I think it would be interesting to provide more information about this. I believe that adding an introductory paragraph explaining the problems that exist in relation to water and why it is important to study alternatives would add a lot of value to the work.
  • In relation to the previous point, this sentence: 'Previous experience in implementing reclaimed water for agricultural irrigation is satisfactory [3]' (lines 2-3), I think it would be more interesting to briefly present the results of this study. Furthermore, it would be much more complete if some more references to studies of this type were added.
  • Lines 60-67 show SUWANU and SuWaNu and I'm not clear on the difference between them. I think that if possible, it should be better explained what the project is about and what is the difference between them.
  • In lines 263-271, I think it is not necessary to explain again what has been done in the work and the groups resulting from the cluster analysis. So, couldn't the comments made between lines 272-305 be distributed when each of the groups is previously discussed? In this way the comments on each of the clusters would be unified.
  • Perhaps it would be interesting to do a search for articles that have been made on the topic analysed in the selected areas and comment on them, to see if they present similar results or contemplate other barriers or ways of dealing with them.

I also suggest the introduction of the following specific suggestions:

  • In table 2 and the appended tables, I think it would be better for visual purposes to separate the 'SWOT dimensions' and the 'Aspects classification' into two columns.
  • Why does line 197 show the WWTP for the first time without indicating what the acronym stands for? It is true that previously it refers to UWWTP and it can be deduced, but it would be better to specify it.
  • On line 88 it says 'Strength, weakness, opportunity and threat', why 'Strength' with a capital letter and not the rest?
  • In line 166 we should put 'an external' and not 'a external'.
  • On line 13 you should put 'results' and not 'result'.

Author Response

Reviewer 3

Dear reviewer, we are pleased to resubmit our article including the suggestions and comments made by the reviewers, whom we thank for their effort and time. The modifications in the text are marked (in red), and below you will find our replies to specific comments (in italics). We hope that this new version is now acceptable.

Thank you very much for your comments. We tried to apply them and improve the paper. However, we must comment you something that happened with reviewer 1 comments. R1 suggested some comments and applying them we found out a mistake in the calculation of the means values. Consequently, we solved the wrongdoing and we remade the analysis.

As a result, the cluster analysis changed and we identified two new clusters: the first one with Belgium, Bulgaria, Germany, Greece and Portugal, and a second one with France, Italy, and Spain. We conducted an in-depth analysis again to understand the structure of these clusters and a new discussion is provided.

In our opinion, results are now more coherent and there appears expected north-south differences. In this case, the Mediterranean and not Mediterranean countries (with the exception of Greece). We did not change nor the methodology, nor the study. Following R1 comments we also add two new analyses to section 3 about the positive-negative SWOT aspects prevalence and the internal-external SWOT prevalence.

In the following lines, we explained how we answered your comments.

Comments and Suggestions for Authors

In this article an analysis is carried out to determine the existing perception of the use of reclaimed wastewater for agricultural irrigation in different regions of the European Union. SWOT analysis and cluster analysis are used for this purpose. In general, the article is well-written, well-organized and all sections are well-developed. The research design is appropriate, and the method is adequately described. The result is clearly presented, and the conclusion are supported by the result. However, the article should introduce some general and specific changes before publishing.

In order to improve the content of this paper the authors should consider following general suggestions:

1] In line 2, when it is mentioned that the use of reclaimed water is often considered a 'win-win' solution, I think it would be interesting to provide more information about this. I believe that adding an introductory paragraph explaining the problems that exist in relation to water and why it is important to study alternatives would add a lot of value to the work.

2] In relation to the previous point, this sentence: 'Previous experience in implementing reclaimed water for agricultural irrigation is satisfactory [3]' (lines 2-3), I think it would be more interesting to briefly present the results of this study. Furthermore, it would be much more complete if some more references to studies of this type were added.

We have included in Section 1 some additional publications that treat the subject (see point [5] below) and also some references that treat the subject and expanded the line 2-3 with the following references and explanation.

'Previous experience in implementing reclaimed water for agricultural irrigation is satisfactory, especially, in water-scarce areas (3) such as Spain, California and Australia (4) Jordan (5- SALAHAT), Italy (6-SAMARAH..), although there are still some barriers and obstacles that have been reviewed by (7-Duong,)

References added:

3. Parsons, L.R.; Sheikh, B.; Holden, R.; York, D.W. Reclaimed water as an alternative water source for crop irrigation. HortScience 2010, 45, 1626–1629, doi:10.21273/hortsci.45.11.1626.

4. Berbel, J.; Esteban, E. Droughts as a catalyst for water policy change. Analysis of Spain, Australia (MDB), and California. Glob. Environ. Chang. 2019, 58, 101969, doi:10.1016/j.gloenvcha.2019.101969.

5. Salahat, M. A., Al-Qinna, M. I., & Badran, R. A. (2017). Potential of treated wastewater usage for adaptation to climate change: Jordan as a success story. In Water Resources in Arid Areas: The Way Forward (pp. 383-405). Springer, Cham.

6. Samarah, N. H., Bashabsheh, K. Y., & Mazahrih, N. T. (2020). Treated wastewater outperformed freshwater for barley irrigation in arid lands. Italian Journal of Agronomy.

7. Duong, K., & Saphores, J. D. M. (2015). Obstacles to wastewater reuse: an overview. Wiley Interdisciplinary Reviews: Water, 2(3), 199-214.

3] Lines 60-67 show SUWANU and SuWaNu and I'm not clear on the difference between them. I think that if possible, it should be better explained what the project is about and what is the difference between them.

This paragraph has been modified and clarified

4] In lines 263-271, I think it is not necessary to explain again what has been done in the work and the groups resulting from the cluster analysis. So, couldn't the comments made between lines 272-305 be distributed when each of the groups is previously discussed? In this way the comments on each of the clusters would be unified.

The discussion section has been modified including the new results and this comment, we hope that it is now adequate.

5] Perhaps it would be interesting to do a search for articles that have been made on the topic analysed in the selected areas and comment on them, to see if they present similar results or contemplate other barriers or ways of dealing with them.

Unfortunately, we have not been able to find paper specifically in the regions under survey. But we have included in Section 1 some additional publications that treat the subject such as:

• European Commission. (2019). Water Fitness Check (SWD(2019) 439). Retrieved from Brussels
• European Commission, (2019) Evaluation of the Council Directive 91/271/EEC of 21 May 1991, concerning urban waste-water treatment. SWD 448-701

6] I also suggest the introduction of the following specific suggestions:

6A] In table 2 and the appended tables, I think it would be better for visual purposes to separate the 'SWOT dimensions' and the 'Aspects classification' into two columns.

Taken this comment and reviewer 1 we add a graphic with a heatmap that makes easier the comprehension of the results. We hope you agree with this, if you considered it is not enough, please let us know

6B] Why does line 197 show the WWTP for the first time without indicating what the acronym stands for? It is true that previously it refers to UWWTP and it can be deduced, but it would be better to specify it.

Thank you for the comment. We have unified the terminology to WWTP

7] On line 88 it says 'Strength, weakness, opportunity and threat', why 'Strength' with a capital letter and not the rest?

Thank you for the comment. We agree with the reviewer and we use capital letters to refer to each element of the analysis

8] In line 166 we should put 'an external' and not 'a external'

This error is corrected.

9] On line 13 you should put 'results' and not 'result'.
This error is corrected.

Round 2

Reviewer 1 Report

Dear editors,

The authors have made a remarkable work in revising their manuscript. The manuscript is clearer and enriched by the recalculation, and the subsequent discussion, they have executed.

I recommend its publication as it stands.

Best,

Reviewer 2 Report

The authors improved the paper, which is now suitable for publication.